# An Adaptive Topology Management Scheme to Maintain Network Connectivity in Wireless Sensor Networks

**DOI:** 10.3390/s22082855

**Published:** 2022-04-08

**Authors:** Muhammad Zia Ul Haq, Muhammad Zahid Khan, Haseeb Ur Rehman, Gulzar Mehmood, Ahmed Binmahfoudh, Moez Krichen, Roobaea Alroobaea

**Affiliations:** 1Department of Electrical and Computer Engineering, Wah Campus, COMSATS University Islamabad, Wah 47040, Pakistan; ziaulhaq_cui@ciitwah.edu.pk; 2Department of Computer Science and Information Technology, University of Malakand, Chakdara 18800, Pakistan; mzahidkhan@uom.edu.pk (M.Z.K.); haseeburrahman@uom.edu.pk (H.U.R.); 3Department of Computer Science, IQRA National University Peshawar, Swat Campus, Swat 19220, Pakistan; 4Department of Computer Engineering, College of Computers and Information Technology, Taif University, P.O. Box 11099, Taif 21944, Saudi Arabia; a.binmahfoudh@tu.edu.sa; 5Faculty of CSIT, Al-Baha University, Al Bahah 65731, Saudi Arabia; moez.krichen@redcad.org; 6ReDCAD Laboratory, University of Sfax, Sfax 3029, Tunisia; 7Department of Computer Science, College of Computers and Information Technology, Taif University, P.O. Box 11099, Taif 21944, Saudi Arabia; r.robai@tu.edu.sa

**Keywords:** WSNs, routing, network lifetime, energy-efficiency

## Abstract

The roots of Wireless Sensor Networks (WSNs) are tracked back to US military developments, and, currently, WSNs have paved their way into a vast domain of civil applications, especially environmental, critical infrastructure, habitat monitoring, etc. In the majority of these applications, WSNs have been deployed to monitor critical and inaccessible terrains; however, due to their unique and resource-constrained nature, WSNs face many design and deployment challenges in these difficult-to-access working environments, including connectivity maintenance, topology management, reliability, etc. However, for WSNs, topology management and connectivity still remain a major concern in WSNs that hampers their operations, with a direct impact on the overall application performance of WSNs. To address this issue, in this paper, we propose a new topology management and connectivity maintenance scheme called a Tolerating Fault and Maintaining Network Connectivity using Array Antenna (ToMaCAA) for WSNs. ToMaCAA is a system designed to adapt to dynamic structures and maintain network connectivity while consuming fewer network resources. Thereafter, we incorporated a Phase Array Antenna into the existing topology management technologies, proving ToMaCAA to be a novel contribution. This new approach allows a node to connect to the farthest node in the network while conserving resources and energy. Moreover, data transmission is restricted to one route, reducing overheads and conserving energy in various other nodes’ idle listening state. For the implementation of ToMaCAA, the MATLAB network simulation platform has been used to test and analyse its performance. The output results were compared with the benchmark schemes, i.e., Disjoint Path Vector (DPV), Adaptive Disjoint Path Vector (ADPV), and Pickup Non-Critical Node Based k-Connectivity (PINC). The performance of ToMaCAA was evaluated based on different performance metrics, i.e., the network lifetime, total number of transmitted messages, and node failure in WSNs. The output results revealed that the ToMaCAA outperformed the DPV, ADPV, and PINC schemes in terms of maintaining network connectivity during link failures and made the network more fault-tolerant and reliable.

## 1. Introduction

Wireless Sensor Networks (WSNs) are distributed networks enabled by significant advances in digital integrated circuitry, wireless transceivers, and communication technologies. Sensors are autonomous smart devices that monitor and sense physical or environmental phenomena in the sensing area [1]. WSNs are important components in manufacturing, transportation, commercial applications, structural security monitoring, and healthcare applications [2]. Moreover, WSNs provide a physical platform for Internet of Things (IoT) systems that connect smart objects to open doors for many other potential applications. Therefore, the new topology management solutions should use it to ensure network connectivity.

Faults and failures are common in WSNs due to their resource-constrained nature and deployment in harsh and hostile environments. Therefore, the intrinsic behaviour and requirements of self-organization and self-configuration for WSNs need to be considered in the design of WSNs. Especially in a dynamic context, the design of a WSN is critical in terms of sensor node deployment, organization, and configuration. As shown in Figure 1, WSN applications have three essential components: sensor nodes, sink/base station, and end-users. The sensor node detects the surrounding environment and sends or forwards the data to the base station (BS). The sink is considered to have unlimited transmission power. A resource-rich node that receives local data and serves the sensor node by providing management services, such as fault detection, fault recovery, data aggregation, connectivity restoration, and so on [3,4]. After data processing, the BS sends the aggregated data from various sensors to the end-user for necessary additional operations.

Sensor nodes are the basic element of WSNs, and they have fewer resources than BS. Furthermore, they have limited available power, bandwidth and memory, processing, and communication capabilities [3]. The application development in WSNs is complicated due to the resource constraints and unique WSN characteristics. In comparison to other wireless and wired networks, WSNs are particularly prone to failure. Due to their inaccessible deployment in rough terrain, sensor maintenance is very difficult. A node death due to energy exhaustion, or node crash, among other things, might cause network faults and failures. Such faults have a major impact on network services and task execution. For example, connection breakage and network isolation are common occurrences that are primarily concerned with topology management [5]. Nodes become untethered and isolated in the network and, as a result, the node remains with unnecessary potential and permanent errors or faults. Consequently, a group of active sensor nodes stops delivering data to the BS, reducing the network lifetime and severely affecting the QoS (Quality of Service). In most cases, the system application’s objectives are not accomplished [6]. Therefore, for topology management, the use of transmission power control for sustaining network connectivity is an important research field, as this approach is more compatible with WSNs because it requires less human and external resources to restore the connectivity of paths.

**Research Contribution**—To begin, we evaluated current relevant literature to gain a better understanding of the concerns and challenges in WSN connectivity and topology management, as well as how existing solutions contributed to the problem. As a result, we divided our connectivity-maintenance strategy into active and passive techniques. This allowed us to focus on the problem from a different perspective. In this paper, our contribution is the design and development of a scheme called “Tolerating fault and Maintaining network Connectivity using Array Antenna” (ToMaCAA), which leveraged the lifetime of WSNs to track and monitor a sensing field for a long time with strong and dynamic connectivity aspects. We employed a Phase Array Antenna (PAA) to focus the transmission power in the required direction and conserve energy from transmission directions other than those required. PAA is an antenna type that is able to form a beam in different directions without moving it physically. The beam formation is controlled using input signal phase shift to the different emitter in the phase array. PAA allows a node to link with the farthest node while using less power than existing methods. As a result, the node’s residual energy is conserved when communicating with reduced transmission power at first, and when the intermediate links fail, a node can interact with a remote node with increased transmission power to re-establish the path through the alternate links. The advantage of transmission in one way is that it decreases overheads and saves energy in idle listening by other nodes. The ToMaCAA scheme was evaluated using the MATLAB network simulation model, and the results were compared to up-to-date schemes, such as Disjoint Path Vector (DPV) [7], Adaptive Disjoint Path Vector (ADPV) [8], and Pickup Non-Critical Node Based k-Connectivity (PINC) [9] by means of performance matrices, such as the network lifetime, amount of transmission messages, and node failure tolerance. The simulation results showed that the ToMaCAA outperformed the DPV, ADPV, and PINC schemes in terms of maintaining network connectivity during link failures, thereby increasing the network lifetime.

The rest of the paper is structured as follows: Section 2 presents a literature review of the most relevant work available in the perspective of this paper. In Section 3, the proposed scheme “***To****lerating fault and **Ma**intaining Network **C**onnectivity using **A**rray **A**ntenna (ToMaCAA)*” is discussed with simulation results and analysis, and Section 4 provides the conclusion and future direction.

## 2. Literature Review

In this section, a literature review of the existing related work is discussed in detail. Furthermore, we highlight the strengths and limitations of the available proposed approaches. The following summary and critical investigation of the available literature focuses on topology management for maintaining connection and its energy-efficiency aspects.

Clustering, Node Discovery, Sleep Cycle Scheduling, Node Placement, Transmission Power Control, Relay, and Node Mobility Control are some of the topology management strategies used in WSNs for connectivity maintenance and energy conservation [5,6]. Transmission power control-based strategies, on the other hand, achieve better network connectivity and an extended network lifetime. Transmission power control systems consider the resource-constrained nature of WSNs. Therefore, they strive to preserve network connectivity with the least amount of energy and resources. In addition, most existing topology management schemes focus solely on energy conservation, with little attention paid to connectivity maintenance. In [10], M. H. Awaad et al. proposed the LEACH-Z (LEACH zones) protocol to divide the network area into zones and prevent the distance node from contesting to be selected as a Cluster Head (CH). Moreover, this helped in energy conservation and increased the network lifetime as compared to the original LEACH.

Kemal Akkaya et al. [11] addressed the issue of network connectivity disruption owing to the breakdown of one or more sensor nodes in the network. They focused on reducing the communication overhead by efficiently observing nodes and identifying those nodes that might interrupt the network’s connectivity. For single-node failure, they scheduled the Partition Detection and Recovery Algorithm (PADRA). Similarly, for multiple instantaneous node failures, they proposed MPADRA. In this scheme, fewer critical mobile nodes come in and substitute failing critical nodes to recover the network connectivity with the fewest possible node/node transfers. However, the coverage holes are not considered. Likewise, PADRA only encounters cut-vertex node failures and ignores numerous common node failures/faults. Additionally, the effectiveness of these techniques should be measured in a real-life setting, as mobile node replacement may not be achievable due to node deployment in dangerous and inaccessible locations where manual node replacement is impractical [8].

Xijun Wang et al. [12] proposed a distributed and localized topology control system, namely the Residual Energy-aware Shortest Path (RESP). This scheme not only helps to balance the energy usage of the nodes, but also offers network connectivity and fault tolerance in WSNs. RESP guarantees k-edge connectivity while trying to keep the weight path to a lower limit. WSN fault tolerance is a critical operational requirement, as WSNs have limited available energy. Therefore, energy-efficient schemes can significantly extend the lifetime of the network.

Sookyoung Lee et al. [13] proposed an algorithm named Connectivity Restoration with Assure Fault-Tolerance (CRAFT), which employed the relay node replacement technique. Their approach restored network connectivity when many collocated node faults occurred at the same time, producing a fault-tolerant topology for upcoming single-node failures. CRAFT deploys the ideal amount of Relay Nodes (RN) in the network while also building a bi-connected inter-partition topology that can withstand a single node failure. CRAFT decreases the inter partition path by utilizing a heuristic approach to deploy the smallest amount of relay nodes needed, resulting in a highly interconnected network structure. However, in a real-world context, the usage of CRAFT is debatable, because sensor nodes are often deployed in holistic locations where replacement is impossible.

Hakki Bagci et al. [3] worked on the issue of network connectivity (sensor-to-sensor and sensor-to-super-node disruption) that happens due to node expiry or isolation, energy depletion, hardware faults, and link faults. The network performance and lifetime are directly affected by this connectivity failure. The Disjoint Path Vector (DPV) algorithm was proposed by the topology control mechanism to solve the problem. This study discussed the problem of topology control in the framework of communication range control while making the system fault-tolerant by employing a static technique to establish k-path disjoint routes from the sensor to a major node. DPV maintains connection with greater fault tolerance, although the energy consumption parameter is not considered. Furthermore, the super node connectivity is affected as a result of the unbalanced energy consumption, affecting network operations and performance.

To solve the problem of energy efficiency in the DPV scheme, F. Deniz et al. [8] proposed an Adaptive Disjoint Path Vector (ADPV) approach. ADPV maintains network connectivity by choosing an alternate path based on the remaining energy of the node providing the path. The selection of the path is carried out by the super node—a resource-rich node. A super node is a resource-rich, heterogeneous node that performs localized management while consuming as few resources as possible. They did, however, employ an omnidirectional antenna and did not define the position of the super nodes, assuming that super node communication is reliable. Since the range of an omnidirectional antenna is smaller than that of a unidirectional antenna, if the antenna is not placed at the centre, it might create interruption and, as a result, damage the coverage of the sensing field. This breakdown of a link’s connectivity has a direct impact on the network’s performance and lifespan. P. Kar et al. [14] presented the Connectivity Reinstatement scheme using Adaptable Sensor Nodes in the existence of Dumb Nodes (CoRAD). In static WSNs, CoRAD emphasized the loss of connectivity caused by intermediate dumb node behaviour. This problem was solved in the past by increasing the transmission power of the neighbour nodes [15], placing a relay node [16] to restore connectivity after a loss, and repositioning the neighbour nodes [17]. It accomplished this by turning on the sleep nodes or altering the communication range of the other nodes. Furthermore, when the node resumed to normal conditions in the adaptive environment, this approach neutralized the extra activated nodes and decreased the communication range of some nodes. CoRAD, on the other hand, focused solely on immediate issues while ignoring long-term and possible flaws. V. K. Akram et al. [9] addressed the issue of network connectivity due to the failure of critical intermediate node and proposed Pickup Non-Critical Node-Based k-Connectivity (PINC) restoration in WSN. The non-critical node physically relocates to the position of the critical failure node with optimum fast movement to restore the network connectivity. However, the relocation of the low-cost sensor node in difficult and cllenging environment is sometimes hard to achieve in such types of WSN application.

After the detailed analysis of the related work, it is evident that these existing schemes use either proactive or reactive approaches to maintain network connectivity, or both, at times. After an intermediate node fails, causing network connectivity to be interrupted, these methods react differently. The RN (Relay Node) placement in the CRAFT [11] method acts as a link between the unconnected portion and the network. Although these topology management approaches are beneficial, manual relay node deployment in a dynamic environment is not always possible. Besides, PADRA and MPADRA [13] used a variety of approaches for repositioning the mobile sensor node in order to reactivate the isolated cut-vertex node. Although this method effectively restores network connectivity, the resource limits and hard deployment settings prevent effective node relocation in WSNs, and moving the sensor node from its first deployed site is problematic in some static networks. Transmission power control, which changes the transmission power to adapt and regulate the transmission range of network nodes, is another potential and suitable option for sustaining network connectivity in topology management. For merging network segments into a single network, DPV [3], ADPV [8], RESP [12], CoRAD [14], CoRD, CoRDWAC [15], and PINC [9] are used.

The detailed literature review confirmed that connectivity maintenance or re-establishment is an important issue in large-scale WSNs and has been extensively investigated [18]. The majority of the work presented in [19] for network connectivity restoration focused on static networks, which have restrictions when used for dynamic WSNs. Furthermore, several other schemes were proposed in [20,21], most of which considered the dynamic features of WSNs. However, such systems have a substantial communication and processing cost for resource-constrained WSNs, making their viability for large-scale WSNs unclear. This research offers intelligent fault detection and energy-efficient fault-tolerance techniques based on reinforcement learning to find the best route with minimum end-to-end delay. Moreover, these systems use omnidirectional antennas, which consume important node energy; nevertheless, most communication requirements are unidirectional in most applications. Furthermore, the topology control system must share data with nearby nodes before executing an algorithm to make topology control decisions. These requirements, on the other hand, raise the network communication overheads and processing complexity. Although wireless communication consumes more energy than any other operation in WSNs [22], the communication overhead is more important than computing complexity.

In WSNs, single link failure or a loss of connectivity has a direct influence on the network’s total performance. The researchers in [23,24] concentrated on the matter of temporary and permanent connectivity loss in static or mobile networks caused by single or many sudden node failures. Such temporary or permanent link loss in connection was recovered, and the network’s core organization and topology were maintained. A. Song et al. tested 30 different WSN topologies and analysed the connection fault and link failures that occurred in WSNs in [25]. They used the equations to undertake a virtual small-world mathematical study to determine the minimal number of nodes required from one end of the network to the other end with the following formulae:[Log_2_N] + 1 ≥ D_min_
where N is the number of nodes, and D is the distance.

Only two topologies out of thirty were found to be appropriate for WSNs. Additionally, WSNs are not yet well connected like social networks. This challenging situation can be solved through statistical models [24]. Mainly, a topology control mechanism is used: “to modify the nodes/network parameters and variate network link modes periodically or after detection of some define events for the aim of increasing the network lifetime while preserving the important characteristic of WSNs like coverage and connectivity etc. [25]”. As demonstrated in Figure 2a below, if the topology control mechanism is developed incorrectly, the scheme will consume more resources than normal. Due to a lack of adequate network topology management, disconnected links and higher transmission power will occur. As a result, precious sensor network resources are wasted, hence shortening the network’s lifetime.

Some points to consider when creating real topology management solutions: in light of the extensive literature survey and analysis, we suggest that the following important design criteria that should be considered when designing and implementing effective topology management solutions for resource-constrained WSNs:For decision-making, distributed data processing should be used.Long-term coverage and connectivity in the occurrence of faults to make the network more fault-tolerant.Algorithms must perform well at the lowest node degree possible.Bidirectional lines should be constructed for reliable connectivity.The scheme would be small (energy efficient) and easy to design and implement.To sustain connectivity, the link recovery process should be rapid.

As a result, we believe that new and effective topology management schemes for resource constrained widespread WSNs are still required.

## 3. Proposed Scheme

In this section, we explain our newly proposed Tolerating fault and Maintaining Network Connectivity using Array Antenna (ToMaCAA) scheme in detail. We first express the motivation and design choices of ToMaCAA and then explain its operation.

### 3.1. Tolerating Faults and Maintaining Network Connectivity Consuming Array Antenna (ToMaCAA)

Application development in WSNs is challenging due to resource constraints and unique WSN characteristics. Compared to other wireless and wired networks, WSNs are notoriously prone to failure. Furthermore, due to their inaccessible deployment in rough terrains, sensor maintenance is uncertain. When a sensor network fails to perform its intended functions, it is seen as a system failure, which could be caused by sensor device problems. A hardware failure, node death due to energy depletion, or node fault, among other things, might cause these problems. These faults have a significant impact on network services and functions; for example, connection breakage and network isolation are common occurrences [26]. Nodes become loose and isolated in the network as a result of unavoidable potential and permanent faults and failures. As a result, a group of active sensor nodes stops delivering data to the BS, reducing the network’s life and severely affecting the QoS (Quality of Service), and, in the vast majority of cases, the objectives of the system applications are not achieved [27,28].

WSNs are significantly more prone to failure than any other kind of wireless network due to resource constraints and hostile deployment. Faults and failures can be caused by a variety of factors, including hardware fragility, a dynamic network environment, unstable wireless media, or a faulty network mechanism. As a result, one of the most important concerns related to network topology management is connectivity maintenance/recovery. Because WSN applications are used in hostile and hazardous conditions where the human administration of nodes and networks is difficult, connectivity restoration in WSNs must be self-organized and self-configured. Various connectivity-restoration strategies have been proposed to overcome this issue, such as the movement of mobile nodes to perform the connectivity function for broken nodes and the establishment of relay nodes to restore communication between isolated regions of the network after nodes expire. Other methods for conserving energy at a node include varying the transmission power for various nodes. So far, these techniques have provided alternate paths to preserve network connectivity, but they consume more energy and reduce network longevity. This study proposed a scheme named distributed and energy-efficient Tolerating faults/errors and Maintaining Network Connectivity using Array Antenna (ToMaCAA) that, unlike traditional WSNs methods, employs a directional array antenna to improve communication efficiency and save energy by avoiding unnecessary transmission.

#### 3.1.1. ToMaCAA: System Design and Operations Overview

The operations of ToMaCAA are divided into two phases from the architectural and system design perspectives: the network initialization phase and the maintenance phase. The network initialization phase is necessary for early formation, whereas the maintenance phase is needed to address a connection breakage to restore connectivity. In the proposed model, there are two types of sensor devices: common sensor and sensor nodes of the topology manager.

**Common Sensor Nodes**—There are ordinary, homogeneous sensor nodes that can sense, process, and communicate. These nodes have limited resources and are prone to failure.

**Topology Manager Nodes**—There are heterogeneous nodes deployed across the network that are resource-rich and have a longer radio range. During the network initialization phase, these nodes initiate network configuration messages. They also start the re-configuration (maintenance) process.

***Assumptions***—We make a number of assumptions in order to provide a consistent system framework for the development of ToMaCAA and its operations. These assumptions are reasonable as a result of technological advances in wireless sensor technology and hardware, and they are as follows:○A group of static and homogeneous common sensor nodes with varying levels of transmission power are placed.○Topology manager nodes with a high transmission range and abundant resources are deployed in a definite number (i.e., 5% of the deployed nodes);○All nodes in the network are approachable from the source node (CS/TM) via their particular power level (connectivity range) and phase angle (direction).

**System Design Choices**—Bagci et al. [2] derived a k-disjoint path from a pool of different paths. Therefore, the transmission range of ADPV (Adaptive Disjoint Path Vector) is lower than that of PAA (Phase Array Antenna) due to the use of an isotropic antenna. As a result, we employed the identical PAA in our design. Furthermore, another inherent benefit of PAA is that similar kinds of communication can be accomplished with low or lowered transmission power. Since the transmission range is enhanced, we may keep fewer nodes “on” to accomplish the necessary network functioning, and the remaining “off” nodes can be utilized for maintenance. Furthermore, as compared to the ADVP and PINC algorithms for a similar node density and communication power, the network size can be increased. The signal strength is boosted when a unidirectional antenna is used.

We focused on the same system or network design that DPV and ADPV evaluated in this study. Our proposed ToMaCAA, on the other hand, achieved reliable connectivity, reduced energy consumption, and longer network lifetime. Furthermore, by employing a unidirectional antenna, the communication overhead is decreased because the transmission is one-sided, conserving both bandwidth and energy. Finally, because the information about the deployed neighbour sensor nodes is saved in the sensor memory nodes through the initialization phase, the amount of update messages in ToMaCAA is minimized. The transmission direction of the PAA [17] used in ToMaCAA is determined by the phase of the incoming signal. As demonstrated in Figure 3, transmission with variable phases or phase shifts entails transmission in separate directions.

The following set of notations and conventions has been utilized throughout this study to explain the phase-wise operations of ToMaCAA, as mentioned in Table 1 below.

#### 3.1.2. ToMaCAA: Phase-Wise Network Operations

We outline the phase-wise procedure of the proposed ToMaCAA scheme in this section. The two phases of ToMaCAA are shown in Figure 4 below:

#### 3.1.3. Initialization Phase

Initialization is the first operation that occurs when sensor nodes are deployed in a sensing field, with the goal of configuring the network into a connected network and topology, as shown in Figure 5. As soon as the network is formed and settled into an appropriate topology—such as a multi-hop hierarchical topology in our example—the network will be prepared for the next possible action, that is, data fusion or data aggregation, once the initialization step is completed. Furthermore, when constructing and structuring the network, the most reliable and low-cost alternate paths are calculated, requiring the smallest amount of available sensor node resources. These data are saved in the internal memory of a node. The following steps are included in the initialization phase:In the first step, the network start-up and configuration message “conf m” is broadcast to nearby Topology Manager (TM) nodes in the first stage;In the second step, every node records the RSSI (Received Signal Strength Indicator) [19] values associated with each phase of the received signal and broadcasts “*Info_m*” to fill in their neighbour list after receiving the message;The third step involves every node, creating a list of neighbour nodes with items arranged from minimum to maximum (low-cost path to high-cost path) and their particular signal phase (ϕ), as indicated in Table 2. Figure 6 shows how to send and receive data from the node closest to you in a given direction.

In the fourth step, links with RRSI values less than or equal to the defined threshold values T_min_ and T_max_ will be discarded by the receiver nodes only if they have at least two neighbours in their list, as illustrated in Figure 6. The red links are disconnected links during the initialization phase to save communication costs, and they have no influence on the neighbour list;Likewise, in the fifth step, the sensor distance node’s distance from the TM node was estimated (based on the RSSI value and a total number of hops). For example, in Figure 6, node No. 7 eliminates all links with neighbour nodes and maintains only one link with node No. 2, as this link is between T_min_ and T_max_ and has the fewest chances of reaching the TM node, as shown in Figure 6;Finally, in step six, every node updates its transmission strength and direction in accordance with the node at the top of the list, and the information tables are updated accordingly. The final controlled neighbour list is shown in Table 3, and the final topology with connected links is shown in Figure 6.

Following the completion of the initialization phase, linkages are formed between the TM nodes and all sensor nodes in the network, and network activities have been started. If a broken link is discovered during network operations, such as when a TM node fails to receive data, the maintenance phase is initiated to re-construct a substitute path for data transmission, making the network fault-tolerant. The maintenance phase is explained in detail in the following sections.

#### 3.1.4. Maintenance Phase

When the Topology Manager (TM) node in the network detects a sensor node failure during data transmission, causing the network to be divided into segments, the preferred approach in the maintenance phase would efficiently re-establish and sustain network connectivity utilizing the available resources. Following are the procedures and activities of the maintenance phase:When a TM identifies lonely nodes and network connectivity disruptions, it starts the maintenance phase by broadcasting a re-formation message (Conf ID) to the WSN. The ID of these nodes from which data are not expected will be included in this message.If the ID of the node receiving the re-configuration message matches the Conf ID, the node whose connection was cracked will re-configure itself, and a new path will be found and organized with the TM. The list of neighbours, as given in Table 2, will be used for this purpose. The procedure of connection re-configuration is depicted in Figure 7. The red isolating node is the one for which the maintenance process starts, and the cracked link is depicted in Figure 7 as a red dotted line;The topology constructed after the isolated node’s link was reconfigured now has a confirmed communication path to the TM node, so the data transmission procedure continues. Figure 7 shows the newly created path as a dotted black line. With the help of an example, the above-described maintenance process for various nodes has been adopted for these installed nodes in the network.

**Isolation of node S1**:

After isolating node S1, the communication of node S5 with the TM node breaks, as shown in Figure 7. When the TM node receives no messages from nodes S1 and S5, the information table becomes as indicated in Table 3, and the reconfiguration message is broadcasted. As a result, S5 will adjust its direction and communication power according to the subsequent top node in the list, as indicated in the topology-organized last neighbour list, as shown in Table 4.

**Isolation of node S2**:

Whenever node S2 fails, disrupting the connectivity of nodes S4 and S7 at the same time, the TM node will broadcast the “Conf ID” re-configuration message after a period of time without receiving the message. The sensor IDs of these nodes are included in the reconfiguration message. These nodes will begin re-configuration after receiving this broadcast message and will select the second top neighbour direction in the list, restoring the alternate link/path, as illustrated in Figure 8.

## 4. Simulation Results and Analysis

In this article, we presented a new scheme called ToMaCAA for resource-constrained WSNs for topology management in WSNs. We discussed the implementation and evaluation of the proposed scheme in this study to demonstrate its efficacy and performance. ToMaCAA was implemented and analysed in MATLAB (R2014b) simulator [29], which is a widely used simulator. The real advantage of using the MATLAB simulator is that it provides a quick way to assess WSN solutions. The MATLAB simulator provides a collaborative environment with many inherent mathematical functions that are both trustworthy and accurate. The most essential property of the MATLAB simulator is its programming capacity, which is very simple to study and use because it supports a variety of user-developed functions [22].

The output results are presented and discussed in this section. After network organisation, communication between nodes begins with maximum power to reach its farthest neighbour, allowing efficient and wise judgments to be made in every situation. Furthermore, if there is no intermediate node between the sensor and the sink node, the sensor may send its data straight to the sink node. However, intermediate sensor nodes S1, S2, and S3 will forward the data of sensor nodes located beyond them i.e., S4, S5, S6, and S7. Sensor node S2 sends its own data to the topology manager (TM) node as well, as shown in Figure 8.

### 4.1. Simulation Parameters and Radio Energy Model

In our simulation, we compared our suggested ToMaCAA technique with the DPV and ADPV protocols to determine its efficiency. CS nodes = 100–500, common nodes, were arbitrarily distributed in a targeted area of (600 × 600) m^2^ on an X, Y plane throughout the simulation. The heterogeneous Topology Manager (TM) nodes regulate the sensor nodes in a dispersed and localized way, while the homogeneous CS nodes detect and monitor the physical environment. TM nodes are (5–10%) larger than CS nodes, and CS nodes are joined with TM nodes, potentially directly or via a neighbour transmitter.

In the proposed ToMaCAA architecture, in every zone, 5% of the heterogeneous nodes were deployed as TM nodes. In a network with a given number of sensor nodes, this is the ideal number of manager/cluster head nodes. The authors of ADPV, DPV, and PINC addressed this issue, and their findings demonstrate that, with various network parameter settings, 5% of nodes in the network that function as management nodes can achieve improved performance, and it works well for evaluating good connectivity. In the proposed network setup, the same parameters were chosen and 5% of the nodes were considered TM nodes. Thus, the full coverage of the network was possible, and the energy load was also balanced.

The placement of topology manager and cooperative sensor nodes in the network was randomly deployed. The WSN was assumed to be static, with all nodes remaining stationary. All nodes exchange information and modify their neighbours list/routing table during the initialization phase after the initial deployment. For information exchange, the neighbour list has one (1) and up to two (2) hop neighbours. In addition, MICAz [23] Motes were employed in our strategy, which feature three levels for adjusting transmission power based on the distance between nodes. Figure 9 shows a snapshot of the MICAz, with 11.1 mA (at −10 dBm), 14.1 mA (at −5 dBm), and 17.4 mA (at 0 dBm) as the three levels. As in DPV and ADPV, the path loss exponent was set to 2.

The size of the packet differs depending on the sensor deployed for various applications and occurrences. However, in our simulation, such application is explored, in which sensor nodes must perceive the surrounding temperature; in such cases, a single-byte payload may be enough. The closing data distribution messages along with packet caption are 6 bytes or 48 bits long when added to the payload. Sensors are assigned IDs at random. The other simulation parameters utilized in the network deployment are listed in Table 5.

The transceiver unit is perhaps the most energy-intensive part of the sensor device because this device might execute 3000 instructions for the expense of transmitting one bit at a distance of 100 m [17]. As a result, for resource-constrained WSNSs, energy-efficient transmission is critical. The power consumption of a transceiver unit is mostly determined by factors such as the antenna type, data rate, communication range, and data transmission modulation schemes. Table 6 shows the radio energy models used in ToMaCAA.

### 4.2. Performance Matrices and Analysis of Simulation Results

We discussed the evaluation and implementation framework in the preceding section. Furthermore, analysis and discussion of the findings of several experiments conducted using the MATLAB simulation are discussed in this section. Our main goal in implementing and simulating the proposed ToMaCAA method is to validate its correctness, superiority, and efficacy over current DPV and ADPV schemes. We conducted various experiments using the network configuration described in the previous section. In particular, in Section 4.2, we considered the most important network performance measures for assessments:Network lifetime.Tolerance of node failure.Amount of transmission messages.

In the experiments, the proposed ToMaCAA was examined with the existing schemes, such as DPV and ADPV, using the parameters indicated above. For evaluation, the MATLAB simulation platform includes full result tracing [30]. When a message is exchanged between the CS nodes or TM nodes, tracing takes place. When the packet is sent, for example, the message is tracked to a log file for subsequent examination. We may receive a wide range of finding possibilities, such as radio model energy dissipation, network energy consumption, network lifetime, position information, and connection breakage, to name a few. Plotting is conducted using the results provided in the log file. Furthermore, the utility tool was used for plotting graphs and charts.

#### 4.2.1. Network Lifetime

WSNs are typically used in remote and inaccessible locations. Because sensor nodes have limited resources and possess a certain amount of energy, their power consumption must be properly monitored. The most important design component is the use of node energy. Furthermore, the lifetime of the network is related to the battery life of each individual node. Existing topology management strategies for maintaining network connectivity use more energy from sensor nodes, reducing the network’s lifetime [26,31,32]. Since we used PAA, which provides direct messages only in a certain track when required, our suggested ToMaCAA strategy maintains network connectivity with an increased network lifetime. This unidirectional transmission saves a large amount of leftover energy in the sensor node.

**Experiment****No. 1**—We carried out the experiment using the simulation framework that we explained in Section 4.2. The experiments were performed for a variety of sensor nodes, ranging from 100 to 500 nodes, considering the node lifetime in minutes. The network was spread over an approximate 600 m × 600 m area. The data were then traced and plotted on the graph as a result of the simulation. Compared to DPV and ADPV, ToMaCAA had a longer network lifetime, as illustrated in Figure 10. For 500 deployed nodes, the suggested ToMaCAA lasted for more than 60 min. It should be noted that we assumed that 5% of the TM nodes were distributed throughout the network.

The ToMaCAA’s improved longevity was mostly attributable to the adoption of Uni-Directional PAA. Because it transmits in only one direction, this antenna has the extra benefit of energy conservation. ToMaCAA also avoids unnecessary links and only creates new ones when the old ones fail. Because the residual energy of a sensor node was not considered for reconfiguration and path formation, the DPV protocol had the shortest network lifetime. Furthermore, ADPV, which is an improved version of DPV, re-established the path adaptively based on the sensor node’s energy. However, PINC relocates nodes that consume extra energy and thus shows the comparatively lower network lifetime than that of ToMaCAA.

#### 4.2.2. Node’s Fault Tolerance

We conducted the experiments using similar network designs and parameters as stated in Section 4.2 to determine how well ToMaCAA adapted to node failure in contrast to DPV, ADPV and PINC. We used the common sensor node (CS) to evaluate performance using this parameter because it disrupted the network connection and could not communicate with the topology manager node. The network was said to be connected if there was a path between each active CS node and the TM node. The maximum number of dead nodes was then calculated, resulting in a break in link/connectivity between the CS and TM nodes.

**Experiment No. 2**—It is clear from the experimental results shown in Figure 11 that, in the DVP, the loss of 5% of the nodes can interrupt network connectivity. Compared to the total number of nodes placed, this is a very small percentage. ToMaCAA, on the other hand, had a high tolerance for node death, as seen in the graph, which shows that the network will be affected if 20–23 percent of nodes die. It could survive more faults than the DPV protocol due to this. Similarly, ADPV and PINC outperformed DPV, although they had a lower fault tolerance than ToMaCAA. This was because the PAA had a longer range than the omnidirectional antenna. The maximum range of a PAA is 130 m, and it can transfer messages over great distances. As a result, it avoids sending extra local messages through the forwarder node and instead delivers messages straight to the farther node. The ToMaCAA, on the other hand, kept the network strongly connected up to the failure of around 93% of sensor nodes when the number of nodes grew to 500. ToMaCAA gained more options to retain connectivity through different channels as the network became more reliable and connected.

#### 4.2.3. Number of Transmission Messages

Another key performance metric we used to evaluate our proposed ToMaCAA method was the number of messages transmitted in the network. Message transmission is a significant element because it determines how much power is required to plan and constitute the finalized topology. This factor considered the fixed rate of the resulting topology during the initialization phase, and, thus, this feature will determine the power proficiency (i.e., the energy necessary for topology creation) of the finalized topology and its convenience.

**Experiment No. 3**—The same network specifications and parameters as previously described were used in the experiment. We set a maximum of 500 common sensor nodes and compared the number of messages exchanged in the network. Figure 12 shows the simulation results, which revealed that ToMaCAA required more messages than DPV, while the PINC algorithm showed a higher number of transmission messages for connectivity restoration with a lower number of connectivity rounds than ToMaCAA, while ToMaCAA connectivity restoration took 90 more rounds than DPV, as shown in Figure 13. The ToMaCAA’s quantity of transmission messages was lower than those of ADPV and PINC, and ToMaCAA’s connectivity restoration rounds were longer than those of ADPV. This demonstrates good connectivity for ToMaCAA for a relatively small number of transmission messages. This means that the sensor node’s energy will be preserved, ultimately extending the network’s lifespan.

## 5. Conclusions and Future Work

Energy efficiency, topology and connectivity maintenance, and fault tolerance are the prime concerns while designing and developing WSNs for application. In this paper, we proposed an energy-efficient adaptive topology-management system called Tolerating Fault and Maintaining Network Connectivity using Array Antenna (ToMaCAA) to sustain network connectivity in resource-constrained WSNs. Due to the failures of nodes and connectivity loss, communication disruption occurs in WSNs. Therefore, we proposed a new technique in this research work to adapt to dynamic structures and maintain network connectivity while utilizing fewer network resources in WSNs. The proposed work was divided into two phases, the network initialization phase and the maintenance phase. The initialization phase was necessary for initial deployment, whereas the maintenance phase was needed to address a connection failure and reconnect connectivity. The simulation of the proposed scheme was conducted and a comparison of ToMaCAA to existing schemes, such as DPV, ADPV and PINC, using specific parameters was carried out. The output results showed that the proposed scheme outperformed the benchmark schemes in terms of energy efficiency and fault tolerance.

In the future, we intend to extend and implement ToMaCAA for mobile sensor networks. For instance, mobile topology manager nodes or mobile sinks can both be used to accomplish mobility. In addition, ToMaCAA’s main focus is on connectivity maintenance and assumes that the network has no coverage gaps. However, it would be interesting to extend ToMaCAA further in order to assess the impact of coverage gaps on the overall network performance. Because WSNs are prone to faults and failures, coverage gaps can arise at any level of network operation.

## Figures and Tables

**Figure 1 sensors-22-02855-f001:**
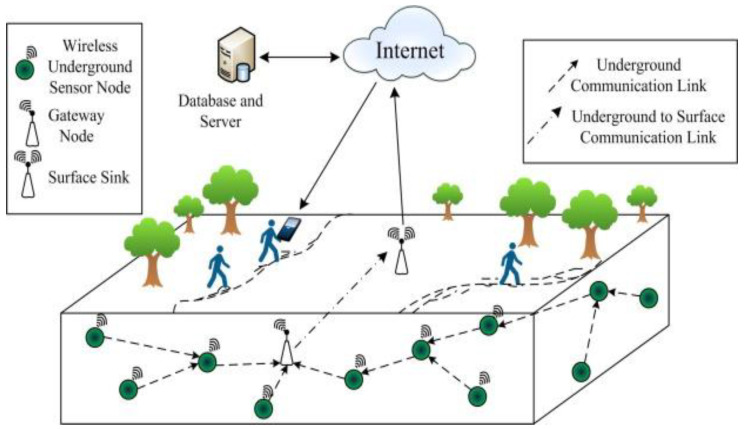
Generic architecture of WSNs.

**Figure 2 sensors-22-02855-f002:**
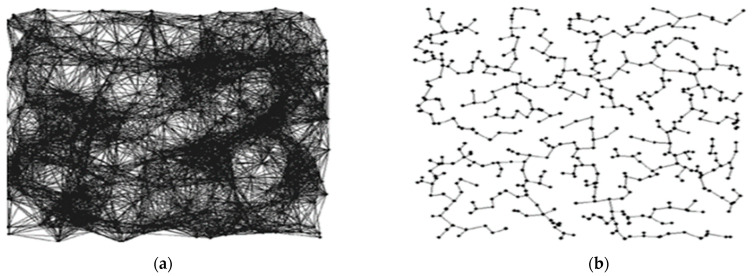
(**a**) The topology at maximum power of a node, (**b**) Reduced topology after applying topology mechanism.

**Figure 3 sensors-22-02855-f003:**
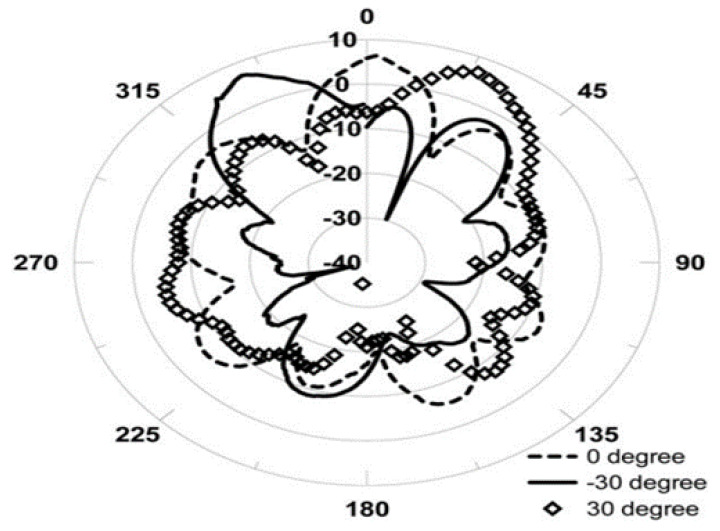
Radiation pattern of the phase array antenna [20].

**Figure 4 sensors-22-02855-f004:**
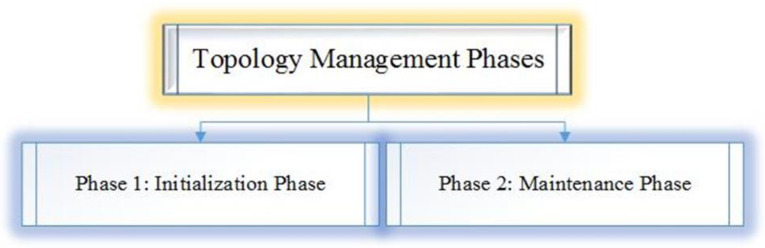
ToMaCAA phase-wise diagram.

**Figure 5 sensors-22-02855-f005:**
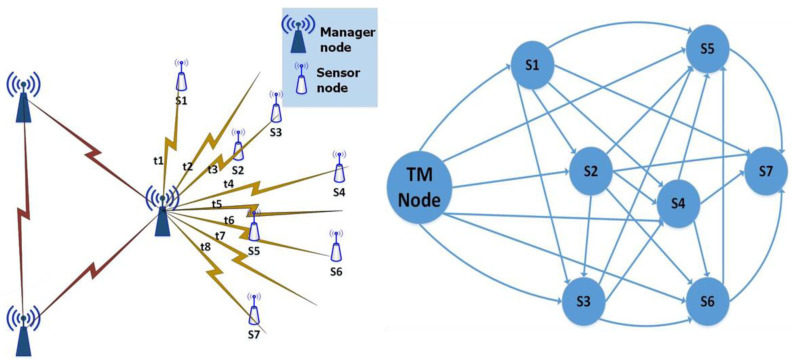
Initial configuration of nodes.

**Figure 6 sensors-22-02855-f006:**
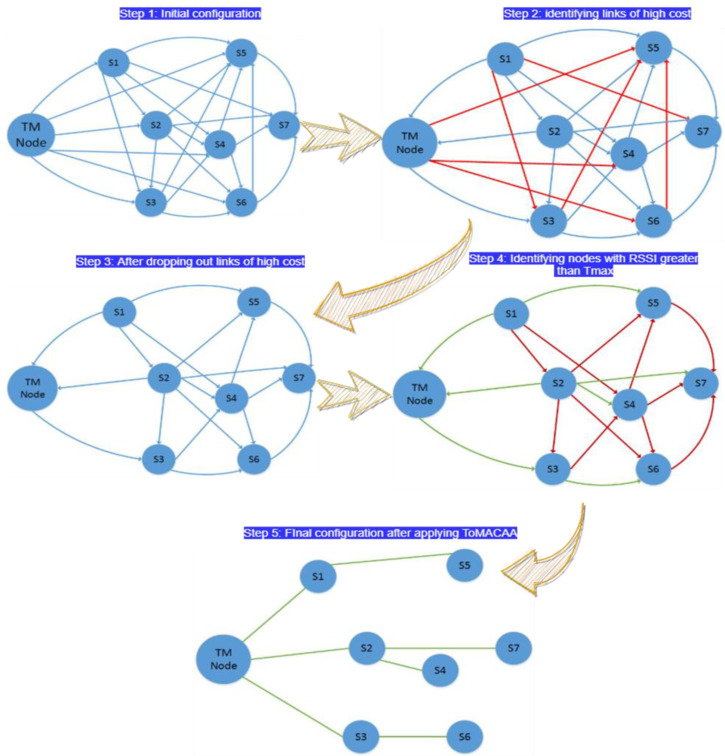
Initialization phase.

**Figure 7 sensors-22-02855-f007:**
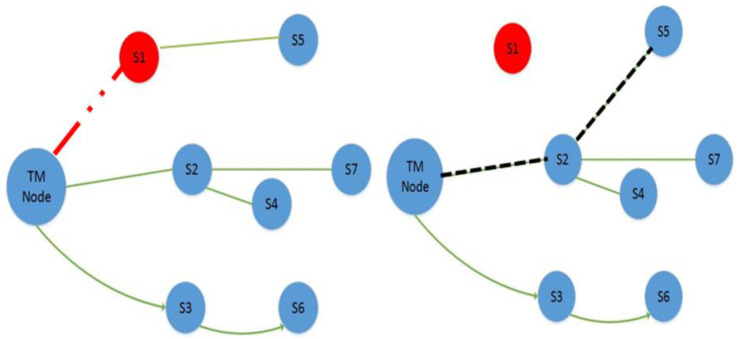
Failure of node S1 and restoration of node S5′s connectivity.

**Figure 8 sensors-22-02855-f008:**
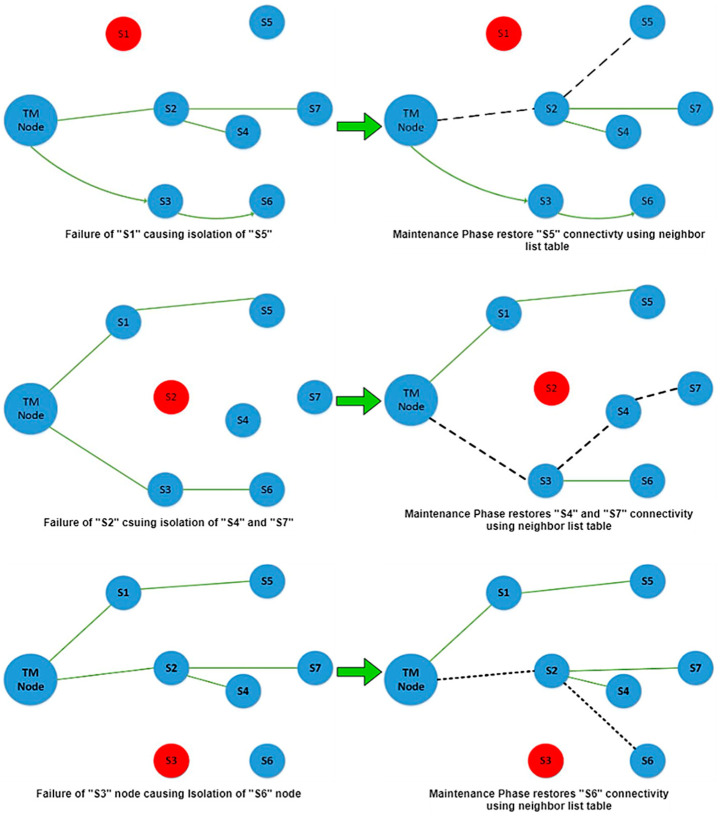
Maintenance phase execution after node failure causes connectivity disruption.

**Figure 9 sensors-22-02855-f009:**
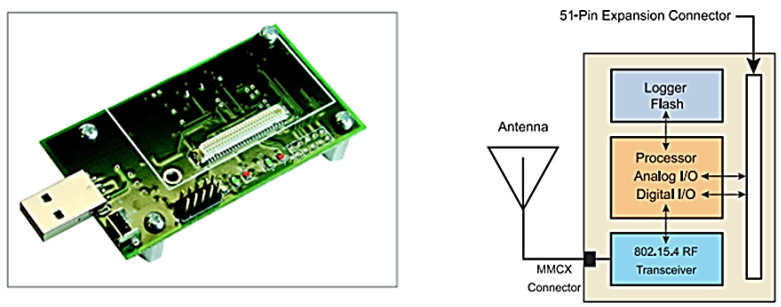
MICAz mote interface board and block diagram of circuitry.

**Figure 10 sensors-22-02855-f010:**
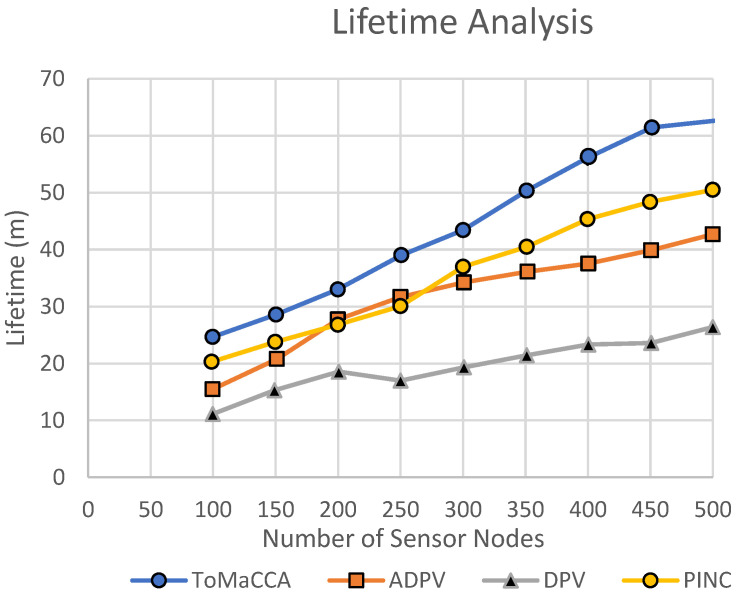
Lifetime evaluation of ToMaCAA with PINC, DPV, and ADPV.

**Figure 11 sensors-22-02855-f011:**
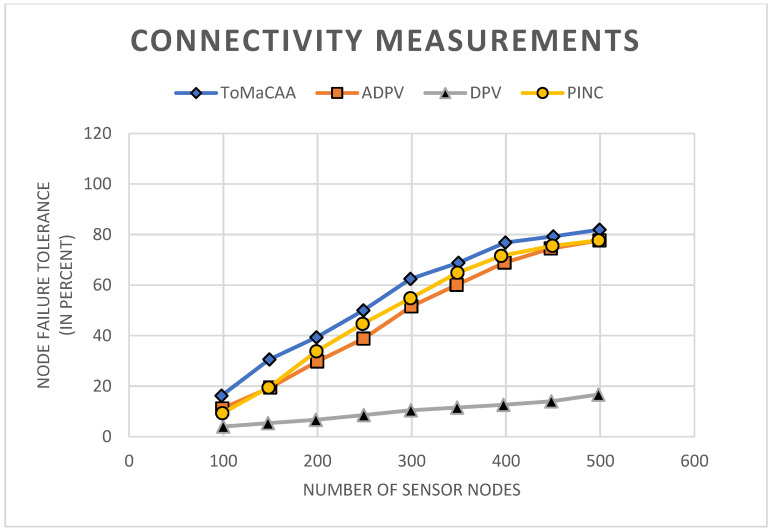
Connectivity disruption threshold as a percentage of failed nodes for various network sizes.

**Figure 12 sensors-22-02855-f012:**
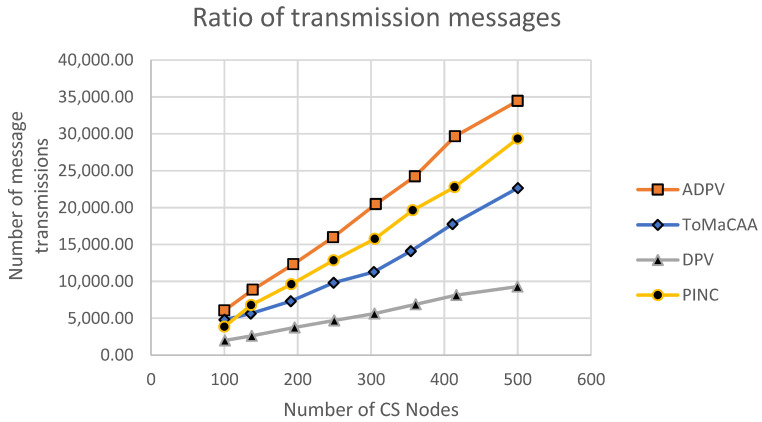
The number of messages transmitted in DPV ADPV, PINC, and ToMaCAA.

**Figure 13 sensors-22-02855-f013:**
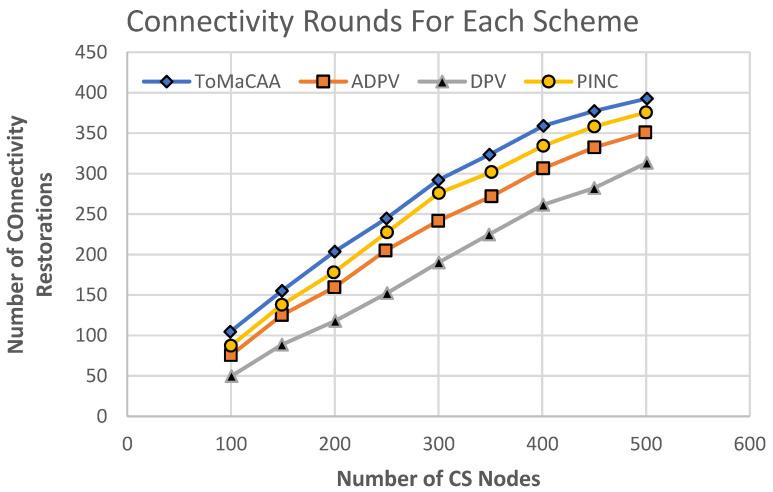
Connectivity restoration in DPV, ADPV, PINC, and ToMaCAA.

**Table 1 sensors-22-02855-t001:** Set of notations.

Terms	Notations
Conf_m	Configuration message
Info_m	Information message (contain node IDs, number of hops from the TM node)
P_t_	Total paths to be stored (maximum of 5)
P_cost_	Path cost
U	Union of two paths
CS	Common sensor node
TM	Topology Manager node
T_min_	Minimum threshold used to drop a link in the initialization phase
T_max_	Maximum threshold used to drop a link in the initialization phase
RSSI	Received signal strength indicator
ϕ	Phase of the signal belonging to the stored link

**Table 2 sensors-22-02855-t002:** Initially generated neighbours list.

**S1 neighbor list**		**S2 neighbor list**		**S3 neighbor list**		**S4 neighbor list**	
TM node	ɸ0	TM node	ɸ0′	TM node	ɸ0″	S2	ɸ1′″
S2	ɸ1	S1	ɸ2′	S2	ɸ1″	S3	ɸ4′″
S3	ɸ4	S3	ɸ3′	S1	ɸ4″	S1	ɸ2′″
S4	ɸ3	S4	ɸ1′	S4	ɸ3″	S6	ɸ3′″
S5	ɸ2	S6	ɸ4′	S6	ɸ2″	S5	ɸ5′″
**S5 neighbor list**			**S6 neighbor list**			**S7 neighbor list**	
S1	ɸ4″″		S3	ɸ2′″″		S2	ɸ4″″″
S2	ɸ3″″		S2	ɸ4′″″		S4	ɸ1″″″
S4	ɸ2″″		S4	ɸ1′″″		S1	ɸ5″″″
S7	ɸ1″″		S5	ɸ5′″″		S6	ɸ3″″″
S6	ɸ5″″		S7	ɸ3′″″		S5	ɸ2″″″

**Table 3 sensors-22-02855-t003:** ToMaCAA controlled final neighbour list.

**S1 neighbor list**		**S2 neighbor list**		**S3 neighbor list**		**S4 neighbor list**	
TM node	ɸ0	TM node	ɸ0′	TM node	ɸ0″	S2	ɸ1′″
**S5 neighbor list**		**S6 neighbor list**		**S7 neighbor list**			
S2	ɸ3″″	S3	ɸ2′″″	S2	ɸ4″″″		

**Table 4 sensors-22-02855-t004:** Alternate route selected for S5 node re-connectivity.

**S1 neighbor list**		**S2 neighbor list**		**S3 neighbor list**		**S4 neighbor list**		**S5 neighbor list**	
TM node	ɸ0	TM node	ɸ0′	TM node	ɸ0″	S2	ɸ1′″	S2	ɸ3″″
**S6 neighbor list**		**S7 neighbor list**							
S3	ɸ2′″″	S2	ɸ4″″″						

**Table 5 sensors-22-02855-t005:** Simulation parameters.

Parameters	Values
Area of deployment	600 m × 600 m
R_max_: Original transmission range of nodes	130 m
CS_n_: Number if CS node	100–500
TM_n_: Number of TM nodes	(5–10) % of CS_n_
K: Degree of disjoint connectivity	5
H: Number of hops for neighbourhood	1, 2
α: Path loss exponent	2
Packet loss Rate	10%

**Table 6 sensors-22-02855-t006:** Radio energy model.

MICAz Mote Radio Model Parameters	Values
Range of transmission power	0–1 mw (−24 dbm–0 dbm)
Range of frequency spectrum	2.4 GHz
MAC protocol used (LR-WPAN)	IEEE 802.15.4 Standard
Max up to EPA (Effective Projected Area) range	130 m
Max up to Omni range	86 m
Consumption of energy in receiving: Rx	19.8 mA
Consumption of energy in transmission: (Tx-1)	11.3 mA @ −10 dBm,
Consumption of energy in transmission: (Tx-2)	14.4 mA @ −5 dBm
Consumption of energy in transmission: (Tx-3)	17.4 mA @ 0 dBm,
Listening: idle mode	20 µA
Sleep mode	1 µA
packet delivery ratio (100 %)	For PPA, up to 80 m and for Omni, up to 60 m.

## Data Availability

Not Applicable.

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
