# Peer review of "An Adaptive Topology Management Scheme to Maintain Network Connectivity in Wireless Sensor Networks"

_sensors, 2022, doi:10.3390/s22082855_

Round 1
Reviewer 1 Report
Required comments are:
- Literature Review Section: Some of the research that addresses reducing energy consumption, network lifetime and connectivity in sensor networks may be useful in this section such as
- Awaad, M. H., & Jebbar, W. A. (2015). Extending the WSN lifetime by Dividing the Network Area into a Specific Zones. I. J. Computer Network and Information Security, 2015, 2, 33-39. DOI: 10.5815/ ijcnis.2015.02.04
- Figures and Tables: Figures and tables are unorganized. Figure 6 used in-text before Figure 3 … etc.
- References list: References should follow the MDPI-Sensors style. For example, Researchers' names should follow the Sensors Journal style (please check all the references). Double quotation should be removed from all reference names. The references list requires moderate scrutiny by the authors.
Author Response
REVIEWER - 01
Reviewer#1 - Concern # 1: Literature Review Section: Some of the research that addresses reducing energy consumption, network lifetime and connectivity in sensor networks may be useful in this section such as
Awaad, M. H., & Jebbar, W. A. (2015). Extending the WSN lifetime by Dividing the Network Area into a Specific Zones. I. J. Computer Network and Information Security, 2015, 2, 33-39. DOI: 10.5815/ ijcnis.2015.02.04
Author response & action: Thanks to the reviewers for mentioning the above important paper.We have reviewed this paper and added the related analysis into the manuscript at lines No. 109-112.
Reviewer#1 - Concern # 2: Figures and Tables: Figures and tables are unorganized. Figure 6 used in-text before Figure 3 … etc.
Author response & Action: We checked and fixed the organization of tables and figures accordingly.
Reviewer#1 - Concern # 3: References list: References should follow the MDPI-Sensors style. For example, Researchers' names should follow the Sensors Journal style (please check all the references). Double quotation should be removed from all reference names. The references list requires moderate scrutiny by the authors..
Author Response & Action: We have taken care to provide the requisite referencing MDPI style. Moreover, the manuscript is also double checked and corrected by the MDPI editing team before the final publishing phase to avoid any mistakes in the style.
Reviewer 2 Report
Authors have proposed an adaptive topology management system, ToMaCAA to support network connectivity in resource-constrained WSNs. The approach has potential.
The authors claim in the beginning (and conclusions even) that the solution is energy efficient. However, no experimental result show this.
The second claim is that the solution is particularly capable to sustain resource-constrained WSNs - again, this part is not particularly proven.
The figures / results are quite simple, not sure what is the statistical significance of the results. A discussion needs ot be included - in particular, rssi is not really the most reliable metric in wireless networks. What happens if there are obtacles. What is the targeted communication protocol?
Author Response
REVIEWER - 02
Concern # 1: The authors claim in the beginning (and conclusions even) that the solution is energy efficient. However, no experimental result show this.
Author response and action: The learned reviewer may be aware that in WSNs, energy efficiency can be measured in terms of many parameters. In our proposed work, we have considered the network lifetime aspect of the network. We have achieved the longer overall network lifetime comparatively as shown and discussed in Figure 10, experiment No. 1. As illustrated, the exended network lifetime is gained by using the Phase array anteena in one required direction with less transmission power instead of using omni directional antennas. In short, we achieved energy efficiency by conserving the transmission power of the sensor nodes. Furthermore, our main focus in this work was on restoring connectivity, and we performed further experiments in that context.
Concern # 2: The second claim is that the solution is particularly capable to sustain resource-constrained WSNs - again, this part is not particularly proven.
Author response and action: Since, WSNs are highly resource constrained, therefore, any solutions devised must be energy efficient in terms of desing and operations. Our proposed solution have proven to be energy efficient and suitable for resource constrained WSNs due to the fact that we have successfully achieved node level fault tolerance by restoring and maintaining connectivity as desired. The proposed ToMaCAA scheme gains more options to retain connectivity through different channels as the network becomes reliable and connected, as discussed in experiment No. 2 and illustrated in graph Figure No. 11. Furthermore, we performed experiments to optimize the overall network lifetime (experiement no. 1) and the number of messages transmitted (experiment no.3) in the network. The experimental results show the better performance of our proposed scheme in comparison to other solutions and support our claim that ToMaCAA is suitiable for resource-constrained WSNs.
Reviewer-02 - Concern # 3: The figures / results are quite simple, not sure what is the statistical significance of the results. A discussion needs ot be included - in particular, rssi is not really the most reliable metric in wireless networks. What happens if there are obtacles. What is the targeted communication protocol?
Author response and action: Firstly, in this work, to the best of our abilities, we performed the simulation based experimental analysis of our proposed ToMaCAA scheme. We conducted various experiments to analyse the network lifetime, fault tolerance and connectivity maintenance, etc., and proved the comparatively better performance of our work. Simulation-based evaluation is widely used in performance analysis of WSNs, however, as suggested by the worthy reviewer, statistical significance is also very important, but due to the time constraints, at this stage more experiments were not possible. However, in future work, we will certainly consider this important aspect of the work to be explored further.
Secondly, the observation about RSSI is interesting. We believe that the RSSI is a cost-effective solution for distance estimation and localization in resource-constrained WSNs. The reason being that, “RSSI does not require any additional hardware at the sensor nodes since the sensor nodes are already equipped with an RF transceiver. It only requires prior knowledge of the estimated path loss factor in the corresponding environment, which can even be estimated offline. However, distance estimation from RSSI measurements suffer from errors due to inaccurate path loss factor estimates and shadowing effects. However, the benefits of achieving a low-cost solution have motivated extensive research to focus on the development of techniques to mitigate the effect of errors in RSSI-based localization schemes ' [1]. Furthermore, many results show that the parameters of the RSSI model are suitable for resource constrained WSNs in estimating the location of target sensors, etc. [2].
Thirdly, the reviewer mentioned that “What if there are obstecles?”. To encounter this issue, in the proposed ToMaCAA scheme, we are able to change the input signal phase and power accordingly, which will change the direction and coverage distance. Hence, the connectivity will be gained through some other direction other than that that has obstacles.
Finally, as we have considered MICAz Mote (sensors), whose operational and communcation standards are predefined just like IEEE 802.15.4 standard. Its enabling operation frequency is 2.4GHZ with 250 kbps data rate while using the DSSS (Direct sequence spread spectrum) used for low-power wireless sensor networks. Details are given in Section 4.1, line no. 425, Figure 9.
[1]. Chuku N, Nasipuri A. RSSI-Based localization schemes for wireless sensor networks using outlier detection. Journal of Sensor and Actuator Networks. 2021 Mar;10(1):10.
[2]. Zheng J, Liu Y, Fan X, Li F. The study of RSSI in wireless sensor networks. In2nd International Conference on Artificial Intelligence and Industrial Engineering (AIIE2016), Advances in Intelligent Systems Research 2016 Nov (Vol. 133, pp. 207-209).
Reviewer 3 Report
First: very minor typos:
line 30: 'proven' should be 'proving'
line 78: I'm not sure what you mean by 'unnecessary potential', could these words be replaced with 'temporary'
line 105: Other than the abstract, this is the first place DPV and ADPV schemes are mentioned so it might be appropriate to spell them out and add the references.
line 160: 'deplation' should be 'depletion'
line 180: Author name should start with a capital (kar -> Kar)
line 188: I think 'decreases their communication' should be 'decreases the communication'
lines 192-196: Based on the context provided by the second sentence you appear to be taking about the physical relocation of a node from one place to another. Is this correct? Could you reword the first sentence to make it clearer. The current wording is a little ambiguous and I was not sure if this meant that the node physically relocates itself to the location of the failed node OR if it meant that the non-critical node assumed the role of the failed node.
line 200: I think this should be reference 9 (not 7)(9 is CRAFT)
line 204: I think this should be reference 7 (not 9)(7 is PADRA)
Line 203: The above two reference error leads to a logical error in the flow of the sentences since PADRA was published in 2010 and CRAFT in 2015. The sentence in line 203 cannot start "As a response, ..." since this paper preceded the one mentioned previously.
line 228: I would imagine that there is one or more really on-point older references that measure and analyse the energy use across all the components of a WSN. (also the wording could be improved, change 'extra' to 'more')
line 249: '... power have been occur' probably should read '... power will occur.'
line 265: 'that a new' should be 'that new' (remove the 'a' as the rest of the sentence is plural)
line 266-267: The sentence "Since transmission power control has proven to be more suitable for WSNs." is missing some context. It doesn't seem to link to the previous sentence.
line 306: There doesn't appear to be a section 3.2 so this subsection should be renumbered 3.1.1 (renumbering should also occur on lines 361, 366, 420)
line 332: Bagci et al. in [2] this is the wrong reference number
line 351: you are talking about the 'phase' of the incoming signal. Are you really talking about the phase or the angle. I would think you were referring to angle.
line 355: Figure 3 - is this the correct reference [18]. I couldn't access the paper referred to but the abstract didn't seem to be antenna orientated (I couldn't tell enough from this abstract)
line 374-375: "This data is saved in the memory of the internal node" did you mean "This data is saved in the internal memory of the node" or were you referring to a specific type of node. This needs clarification.
line 377 onward: you mention six steps but the word for step is inconsistent. You have used step, stage and phase. Please change them all to step, and the third step needs extra adjustment (In the third stage involves) should become (The third step involves) (get rid of 'In').
line 421: I believe that Manage should be Manager.
line 424-425: This sentence is awkward "Following is the procedure how the maintenance phase-wise activities:" Would it be better if it said something like "Following are the procedures and activities of the maintenance phase."
line 468: reference [20]. Is this the correct reference for Matlab simulator?
line 478-479: This sentence has two issues "However, as seen in, an intermediate sensor node will forward the data of sensor-nodes in the back." I'm thinking that you meant to have a reference ... as seen in [n], and maybe the last word should have been 'background' not 'back' unless 'back was talking about more-distant nodes.
line 482: There is no section 4.1 so this section should be renumbered to 4.1 (as should the following sections and subsections).
line 498: should the last words be "is also balanced" ?
line 501: modify not modifies
line 503: hop not hope
line 506: "(at 0dBm)" should immediately follow 17.4mA
line 518: I think "energy eating" should be "energy intensive"
line 520: Is this the right reference. I cannot see any energy cost of CPU cycles compared to Tx energy cost in this ref.
line 525: Table six contains the parameter "max up-to EPA range". What is EPA?
line 527: This heading should be renumbered (as should its subheadings). The heading also needs to be reworded: "4.2 Performance Matrices and Analysis of Simulation Results"
lines 618, 619: Capitialise the word 'figure' (might be others, do a search)
line 643-646: Some spelling and tense errors; "Experiment of the proposed scheme is conduced and comparison of ToMaCAA to existing schemes such as DPV, ADPV and PINC using the parameters are carried out. The output results show that the proposed scheme outperforms the benchmark schemes in terms of efficiency."
Maybe this would be better:
Simulations of the proposed scheme were conducted and comparisons of ToMaCAA to existing schemes such as DPV, ADPV and PINC using specific parameters were carried out. The output results showed that the proposed scheme outperformed the benchmark schemes in terms of energy efficiency and fault tolerance."
********** Placement of Abbreviations ****************
Various places: weird locations of abbreviations in relation to the explanation of the abbreviation, Eg:
Line 334 says "... PAA due to the isotropic antenna (Phase Array Antenna)"
The location of "(Phase Array Antenna)" is weird (and it is unnecessary since PAA was defined back on line 94). (This goes for other places where the abbreviation and explanation are repeated)(lines 350, 555, 571)
Line 480: (TMN) appears near the end of the line when it should immediately follow topology manager node (which should be capitialised)
************** Figures 10-13 **********************************
Can you change the marker styles so the lines can be distinguished in a black & white print of the paper.
Caption for Figure 11 is incomprehensible. Maybe something like "Connectivity Disruption Threshold as a Percentage of Failed Nodes for Various Network Sizes"
************* Experiment vs Simulation ************
Pages 16, 17 & 18 refer to 'Experiment' or 'Experiments' but I believe these are all simulations* (see heading of section 4) and should read "Simulation No.1" etc. (Along with the subsequent changes of 'experiment' to 'simulation' as necessary, do a search so you can make the language consistent)
*as I understand the paper these are simulations run in Matlab, not experiments run on the target systems. In which case, part of your further work would be implementing ToMaCAA on a target network, monitoring it and comparing it to your simulation. (Your further work section may say this but not clearly.)
************************** ISSUE *****************************
line 332 and onward: System Design Choices
The Phase Array Antenna (PAA) is introduced in line 93 but not mentioned again until this section. It is introduced with no background and the first information about it appears in line 350 along with reference [17]. As far as I can tell reference [17] doesn't mention the antenna anywhere.
I am not sure what you are talking about and believe you are referring to a 'Phased Array Antenna' or an electronically steerable antenna where the phase shift applied to various elements of the antenna has the effect of creating the main beam in the desired orientation and conversely, looking at the phase of the incoming signal across the different elements can give the angle to the transmitter.
I am guessing that you are talking about a passive phased array since figure 9 just shows a single transceiver. I would image that there must be some other interface to the antenna to get the angle information
Is it a module? What is the energy cost of steering the antenna compared to the RF gain in the required direction. Are you only expecting nodes to be in a particular direction or can you steer the antenna through 360 degrees? (Figure 3 shows +/- 30 degrees)(Can Figure 3 be modified to show the Radiation Pattern for the default omni-directional antenna for the same input power as a comparison)
What happens when you receive a signal, does it trigger antenna scanning to extract the signal phase and then the sensor cpu calculates the angle?
In Summary: the mention of the PAA lacks any definition and useful information and since the whole paper is about ToMaCAA the Antenna Array needs more information.
************ email addresses *******************
Lines 6,7,8,9: Do you need these email addresses? You have given the correspondence email address so I think the others can be removed.
************ References ************************
Reference 22 doesn't appear to be used in the text.
A number of the references appear out of order and I don't trust that some of the references are the correct number.
References out of order - in the current layout reference 27 should be reference [5]. Other out-of-order references include 21, 24 25, 28 which should be higher up the list. Some references should be lower on the list. Renumbering shouldn't occur until other reference related issues are updated (eg see line 105 comment and other comments)
Author Response
Concern # 1:***********First: very minor typos****************
Author response and actions: We highly appreciate the reviewer’s insightful and helpful comments on our mansucripti.
- To the best of our abilities, we took great precaustion and carefully proofread the menuscrip to remove any possible corrections and typos.
- Many sentences of the manuscript have been carefully rewritten or re-organized to enhance the logic flow and make the statements clearer in tone.
- All the typose, spelling mistakes, grammatical errors, numbering issues, missing abbreviations are reviewed and correted as mentioned by the reviewer. The final manuscript is thoroughly edited, and all the changes in the text has been highlighted at various places in the manuscript.
Concern # 2: ********** Placement of Abbreviations ****************
Various places: weird locations of abbreviations in relation to the explanation of the abbreviation, Eg: Line 334 says "... PAA due to the isotropic antenna (Phase Array Antenna)"
The location of "(Phase Array Antenna)" is weird (and it is unnecessary since PAA was defined back on line 94). (This goes for other places where the abbreviation and explanation are repeated)(lines 350, 555, 571)
Line 480: (TMN) appears near the end of the line when it should immediately follow topology manager node (which should be capitialised)
Author response & action: We wish to express our appreciation for the in-depth comments, suggestions, and corrections that helped improve the manuscript. The miss placed abbreviations have been fixed and relocated to suitable places in the text. The changes have been highlithed in various places.
Concern # 3: ************** Figures 10-13 **********************************
Can you change the marker styles so the lines can be distinguished in a black & white print of the paper.
Caption for Figure 11 is incomprehensible. Maybe something like "Connectivity Disruption Threshold as a Percentage of Failed Nodes for Various Network Sizes"
Author response & action: Suggested corrections are done in the revised manuscript now, and are highlited. Figure 11, line no. 506.
Concern # 4: ************* Experiment vs Simulation ************
Pages 16, 17 & 18 refer to 'Experiment' or 'Experiments' but I believe these are all simulations* (see heading of section 4) and should read "Simulation No.1" etc. (Along with the subsequent changes of 'experiment' to 'simulation' as necessary, do a search so you can make the language consistent)
*as I understand the paper these are simulations run in Matlab, not experiments run on the target systems. In which case, part of your further work would be implementing ToMaCAA on a target network, monitoring it and comparing it to your simulation. (Your further work section may say this but not clearly.)
Author response & action: Righly said that these are simulations done in MATLAB and performance comparison was done. Our humble understanding was that both terms convey similar meanings. We have explained in Section 4.2 Performance Matrics and Analysis of Simulation Results that for analysis different experiments or simulations are performed. We have used the same convention in some of our other published work, and also find the same pattern in many theseses and papers. Therefore, we prefer to stick to this style.
Concern # 5: line 332 and onward: System Design Choices
The Phase Array Antenna (PAA) is introduced in line 93 but not mentioned again until this section. It is introduced with no background and the first information about it appears in line 350 along with reference [17]. As far as I can tell reference [17] doesn't mention the antenna anywhere.
I am not sure what you are talking about and believe you are referring to a 'Phased Array Antenna' or an electronically steerable antenna where the phase shift applied to various elements of the antenna has the effect of creating the main beam in the desired orientation and conversely, looking at the phase of the incoming signal across the different elements can give the angle to the transmitter.
I am guessing that you are talking about a passive phased array since figure 9 just shows a single transceiver. I would image that there must be some other interface to the antenna to get the angle information
Is it a module? What is the energy cost of steering the antenna compared to the RF gain in the required direction. Are you only expecting nodes to be in a particular direction or can you steer the antenna through 360 degrees? (Figure 3 shows +/- 30 degrees)(Can Figure 3 be modified to show the Radiation Pattern for the default omni-directional antenna for the same input power as a comparison)
What happens when you receive a signal, does it trigger antenna scanning to extract the signal phase and then the sensor cpu calculates the angle?
In Summary: the mention of the PAA lacks any definition and useful information and since the whole paper is about ToMaCAA the Antenna Array needs more information.
Author response & action: Thanks for the reviewes feedback on this issue. A PAA definition and necessary details have been added in line 93. Reference issue is also fixed.
We have used a PAA (not an electronically steerable antenna) because it is more feasible in WSN as it is less costly, energy efficient and can be deployed in hostile environments. The direction of the output beam of PAA quietly depends on the phases of the input signal for various elements of PAA. Furthermore, Figure 3 shows the PAA transmission pattern, showing only three beams in different directions with different input phase degrees. However, the input phase will be calculated in node internal processing module using received signal phase that may be different from figure no.3 and may be steared (lembda/2)=90o. When the signal is received from the neighbour node, the info_m message consists of input signal phase from source and RSSI at the receiving node, then the sensor node CPU updates the neighbor node table with calculated phase values.
Concern # 6: ************ email addresses *******************
Lines 6,7,8,9: Do you need these email addresses? You have given the correspondence email address so I think the others can be removed.
Author response & action: Corrections done and unnecessary details are removed.
Concern # 7: ************ References ************************
Reference 22 doesn't appear to be used in the text.
A number of the references appear out of order and I don't trust that some of the references are the correct number.
References out of order - in the current layout reference 27 should be reference [5]. Other out-of-order references include 21, 24 25, 28 which should be higher up the list. Some references should be lower on the list. Renumbering shouldn't occur until other reference related issues are updated (eg see line 105 comment and other comments)
Author response & action: We highly regard and appreciate the keen observations of a worthy reviewer in this regard. The mixup of the numbering of references was due to the use of the different versions of Endnote software by the Authors during the revision. We have now carefully sorted out the issue and corrected the references accordingly and made sure that all citations are in line with references in the manuscript.
We really hope that the revised text has clarified our changes sufficiently for the reviewers to accept them. Thanks. During the peer review process, we appreciate your assistance in improving the readability and quality of our paper.
Round 2
Reviewer 2 Report
I stick to my initial comments regarding the interpretation of statistical information in the obtained results.
Author Response
First of all, initially, we did not find out the statistical significance tests of our results, because, none of the related work in the literature review we discussed, mentioned or tested the statistical significance. However, we agree with your suggestion that adding the statistical significance would certainly be of interest, and can further improve the evaluation and analysis.
Secondly, the work submitted in this article was part of the research undertaken in the MPhil study. The project has been concluded, and due to the closure of the project and contract termination, the manpower required for this work is not available.
Moreover, unfortunately, much to our regret, the researcher and implementers of this work are engaged in other projects and Ph.D. studies, and it is not possible for us to perform further experiments and tests.
Therefore, we humbly request that the condition of test for statistical significance may please be waived, and accept the revised article for publication. Nevertheless, if the next improved version of this paper with further experiments and evaluation is submitted in the future, then we will positively consider the statistical significance aspect of our study.
This manuscript is a resubmission of an earlier submission. The following is a list of the peer review reports and author responses from that submission.
Round 1
Reviewer 1 Report
Fault-tolerant topology management and routing in sensor networks has been extensively studied. Therefore, a comprehensive literature survey and performance comparison with recent schemes are essential in this topic. Unfortunately, this paper is lack of both. Below are some of related references. The authors have to include a review and performance comparison with them.
Multi-objective fractional gravitational search algorithm for energy efficient routing in IoT
Wireless networks, 2019 - Springer
Energy-aware distributed routing algorithm to tolerate network failure in wireless sensor networks
P Chanak, I Banerjee, RS Sherratt - Ad Hoc Networks, 2017 - Elsevier
An energy efficient and QoS aware routing algorithm based on data classification for industrial wireless sensor networks
W Zhang, Y Liu, G Han, Y Feng, Y Zhao - IEEE Access, 2018
CAMP: cluster aided multi-path routing protocol for wireless sensor networks
M Sajwan, D Gosain, AK Sharma - Wireless Networks, 2019
Invulnerability of clustering wireless sensor networks against cascading failures
X Fu, Y Yang, O Postolache - IEEE Systems Journal, 2018
Distributed k-connectivity restoration for fault tolerant wireless sensor and actuator networks: algorithm design and experimental evaluations
VK Akram, O Dagdeviren, B Tavli - IEEE Transactions on …, 2020
PINC: Pickup Non-Critical Node Based k-Connectivity Restoration in Wireless Sensor Networks
V Khalilpour Akram, Z Akusta Dagdeviren… - Sensors, 2021
Since the main gain of the research comes from the use of antenna array, the authors also have to compare theirs with the works on beamforming/directional antenna in WSNs. Below are some of them:
A Stochastic Beamforming Algorithm for Wireless Sensor Network with Multiple Relays and Multiple Eavesdroppers
Z Hu, Y Jin, H Liu - Wireless Personal Communications, 2021
Review on directional antenna for wireless sensor network applications
R George, TAJ Mary - IET Communications, 2020
Improving Sensor Network Performance with Directional Antennas: A Cross-layer Optimization
J Schandy, S Olofsson, N Gammarano… - … on Sensor Networks …, 2021
It seems some figures are borrowed somewhere without courtesy.
What is \pi in the figures? It has to be defined before its use.
The figures have to be replaced with their high-resolution versions.
Author Response
REVIEWER - 01
Reviewer#1 - Concern # 1: Fault-tolerant topology management and routing in sensor networks has been extensively studied. Therefore, a comprehensive literature survey and performance comparison with recent schemes are essential in this topic. Unfortunately, this paper is lack of both. Below are some of related references. The authors have to include a review and performance comparison with them.
- Multi-objective fractional gravitational search algorithm for energy efficient routing in IoT Wireless networks, 2019 - Springer
- Energy-aware distributed routing algorithm to tolerate network failure in wireless sensor networks, P Chanak, I Banerjee, RS Sherratt - Ad Hoc Networks, 2017 - Elsevier
- An energy efficient and QoS aware routing algorithm based on data classification for industrial wireless sensor networks. W Zhang, Y Liu, G Han, Y Feng, Y Zhao - IEEE Access, 2018
- CAMP: cluster aided multi-path routing protocol for wireless sensor networks, M Sajwan, D Gosain, AK Sharma - Wireless Networks, 2019
- Invulnerability of clustering wireless sensor networks against cascading failures, X Fu, Y Yang, O Postolache - IEEE Systems Journal, 2018
- Distributed k-connectivity restoration for fault tolerant wireless sensor and actuator networks: algorithm design and experimental evaluations, VK Akram, O Dagdeviren, B Tavli - IEEE Transactions on …, 2020
- PINC: Pickup Non-Critical Node Based k-Connectivity Restoration in Wireless Sensor Networks, V Khalilpour Akram, Z Akusta Dagdeviren… - Sensors, 2021
Author response: We are grateful for this suggestion. As suggested by the esteemed reviewer, we have included some of the most relevant references. However, due to the time constraints at this stage, it is not possible to perform experiments and comparisons with new schemes, as this project has been completed. This work was done as MS research and thesis on the same has been defended already. However, in future, it will be interesting to compare it with the references suggested by the reviewer.
Author action: The following four papers are included as per recommendations:
- Multi-objective fractional gravitational search algorithm for energy efficient routing in IoT Wireless networks, 2019 - Springer
- Energy-aware distributed routing algorithm to tolerate network failure in wireless sensor networks, P Chanak, I Banerjee, RS Sherratt - Ad Hoc Networks, 2017 – Elsevier
- CAMP: cluster aided multi-path routing protocol for wireless sensor networks, M Sajwan, D Gosain, AK Sharma - Wireless Networks, 2019
- PINC: Pickup Non-Critical Node Based k-Connectivity Restoration in Wireless Sensor Networks, V Khalilpour Akram, Z Akusta Dagdeviren… - Sensors, 2021
Reviewer#1 - Concern # 2: Since the main gain of the research comes from the use of antenna array, the authors also have to compare theirs with the works on beamforming/directional antenna in WSNs. Below are some of them:
· A Stochastic Beamforming Algorithm for Wireless Sensor Network with Multiple Relays and Multiple Eavesdroppers
· Z Hu, Y Jin, H Liu - Wireless Personal Communications, 2021
· Review on directional antenna for wireless sensor network applications
· R George, TAJ Mary - IET Communications, 2020
· Improving Sensor Network Performance with Directional Antennas: A Cross-layer Optimization
· J Schandy, S Olofsson, N Gammarano… - … on Sensor Networks …, 2021
Author response: Thanks for the reviewer’s comments, and the papers suggested have good proposed work, and they can be considered when in future we want to extend our work to deal with the array antenna and beam forming. While our present research work is related to using array antenna as a technique for achieving long-lasting connectivity only.
Author action: Since our work is mainly concerned with connectivity and comparison with the related schemes has already been done and presented. Therefore, a comparison with the schemes suggested by the reviewer can be considered in future.
Reviewer#1 - Concern # 3: It seems some figures are borrowed somewhere without courtesy.
Author response: Thanks for the reviewer’s comments. Correction done.
Author action: The figure has now been cited/referenced as Figure 1 reference No. [27] and highlighted.
Reviewer#1 - Concern # 4: What is \pi in the figures? It has to be defined before its use.
Author response: Thanks for the reviewer’s comments. Correction done.
Author action: Pi () has been defined in Table 1.
Reviewer#1 - Concern # 5: The figures have to be replaced with their high-resolution versions.
Author response: Thanks for the reviewer’s comments. Correction done.
Author action: Figure 5 has been replaced with a high resolution picture.
Reviewer 2 Report
This research studies a very important topic in the field of networks, especially sensor networks. In this paper, the authors focused on the issue of connectivity in wireless sensor networks. This issue is very important in managing the network properly. The authors propose a ToMaCAA scheme for maintaining network connectivity through incorporating phase array antenna into topology management technologies. Also, they compared their proposed (ToMaCAA) with DPV and ADPV schemes during experiments. This research adds valuable information and appropriately structured the research. However, the authors have to address all of the below concerns carefully.
- Abstract: It is very long, authors should reduce the number of words while maintaining the scientific and sequence of sentences. Also, they should focus on the coherence of the sentences so that the abstract is clearer.
- Literature Review Section: Why some words that written in italic font such as (Clustering, Node Discovery, Sleep Cycle Scheduling … etc. (page3-line11-12))? It should be removed. The word "this" (page3-line125) is not clear, is it related to the research [7] or to the proposed research? We believe the sentences from "analysis, we suggest that the following design important criteria that should" (page6-line239) to "solutions should use it to ensure network connectivity." (page6-line253) should be moved to another section.
- Methodology Section: What is “k-disjoint” (page7-line315)? Research should be comprehensive.
- Simulation Results and Analysis Section: The authors note that "Figure 12 shows the simulation results, which reveal that ToMaCAA requires more messages than DPV." (page18-line583) but they did not explain why. The authors adequately explained the results and trials but did not discuss these results in a separate section. The authors compared their results with DPV and ADPV but did not compare their results with existing research. What are the drawbacks and limitations of the proposed research?
- Figures and Tables: Some figures should be redrawn to be high resolution and to remove blurring such as Figures 1 and 5. Also, Figure 2 shown before being summoned in-text. Figure 5 is not used in-text. Figure 4 used in-text before Figure 3. What is the point of using some colors in tables and figures? Table 1 should be added as a table and not as an image. Figure 4 should minimize it. Why not rely on one color and one format to represent the proposed research in Figures 10 and 11?
- English Writing: This paper requires extensive proofreading. There are some of grammatical, spelling and typos problems. The authors have to thoroughly scrutinize the paper. Without professional, accurate and clear English, readers cannot understand the research.
- References list: References should follow the MDPI-Sensors style. For example, Researchers' names should follow the Sensors Journal style (please check all the references). The double quotation should be removed from all reference names. Some search names in the reference list begin an uppercase letter for each word (such as [3], [6] ... etc.) and others use only an uppercase letter in the first word (such as [1], [2] … etc.), the author should standardize style. The number of references is insufficient for this study. The references list requires extensive scrutiny by the authors.
Author Response
REVIEWER - 02
- Reviewer#2 - Concern # 1: This research studies a very important topic in the field of networks, especially sensor networks. In this paper, the authors focused on the issue of connectivity in wireless sensor networks. This issue is very important in managing the network properly. The authors propose a ToMaCAA scheme for maintaining network connectivity through incorporating phase array antenna into topology management technologies. Also, they compared their proposed (ToMaCAA) with DPV and ADPV schemes during experiments. This research adds valuable information and appropriately structured the research. However, the authors have to address all of the below concerns carefully.
Author response: Thanks for the encouraging comments.
Author action: The suggestions/concerns suggested by the reviewers are considered and corrections are done accordingly.
Reviewer#2 - Concern # 2: Abstract: It is very long, authors should reduce the number of words while maintaining the scientific and sequence of sentences. Also, they should focus on the coherence of the sentences so that the abstract is clearer.
Author response: Thanks for the recommendation.
Author action: Abstract has been revised to bring more clarity, and the changed text is highlighted.
Reviewer#2 - Concern # 3: Literature Review Section: Why some words that written in italic font such as (Clustering, Node Discovery, Sleep Cycle Scheduling … etc. (page3-line11-12))? It should be removed. The word "this" (page3-line125) is not clear, is it related to the research [7] or to the proposed research? We believe the sentences from "analysis, we suggest that the following design important criteria that should" (page6-line239) to "solutions should use it to ensure network connectivity." (page6-line253) should be moved to another section.
Author response: Thanks for the recommendation. We made some notations in Italic only to bring emphasis. However, as suggested we have done the corrections.
Author action: Corrections in Literature Review section is done and highlighted.
Reviewer#2 - Concern # 4: Methodology Section: What is “k-disjoint” (page7-line315)? Research should be comprehensive. .
Author response: Thanks for the recommendation, an explanation about k-disjoint is added.
Author action: K-disjoint paths mean, determining “k” number of possible connected path from CS with neighbor nodes reaching to the TM nodes.
Reviewer#2 - Concern # 5: Simulation Results and Analysis Section: The authors note that "Figure 12 shows the simulation results, which reveal that ToMaCAA requires more messages than DPV." (page18-line583) but they did not explain why. The authors adequately explained the results and trials but did not discuss these results in a separate section. The authors compared their results with DPV and ADPV but did not compare their results with existing research. What are the drawbacks and limitations of the proposed research?
Author response: Thanks for the recommendation. ToMaCAA is actually designed for static wireless sensor networks where sensor nodes are deployed randomly and node failure causes connectivity disruption and loss of information. In section 4.3.3 we explained our simulation results that ToMaCAA needs more overhead messages at initial configuration than DPV and less than ADPV with higher recovery benefits than DPV and ADPV which is the most important aspect of this research. Furthermore, the limitations and drawbacks of the proposed work are mentioned.
Author action: ToMaCAA lakes dealing with coverage holes problem in WSNs, supporting static nodes (sensor and topology manager node) are mentioned in last section. However, one more limitation of the scheme is added with last paragraph and highlighted.
“Because WSNs are prone to faults and failures, and coverage gaps can arise at any level of network operations. ToMaCAA need to be checked for scalability because low-cost sensors are deployed for all applications of WSNs, where these nodes are prone to fault and failure. Therefore, WSN should be scalable, and self-organized to recover from connectivity breakdown with less resource’s utilization”.
Reviewer#2 - Concern # 6: Figures and Tables: Some figures should be redrawn to be high resolution and to remove blurring such as Figures 1 and 5. Also, Figure 2 shown before being summoned in-text. Figure 5 is not used in-text. Figure 4 used in-text before Figure 3. What is the point of using some colors in tables and figures? Table 1 should be added as a table and not as an image. Figure 4 should minimize it. Why not rely on one color and one format to represent the proposed research in Figures 10 and 11?
Author response: Thanks for the recommendation. Figures are now fixed and redrawn for better quality. Corrections are done accordingly.
Author action: Figure 2, 4 and 5 has been shifted now. Color schemes of figures and tables now fixed. Figure 4 size has been fixed now. Table 1 is now added as recommended. Finally, color and format of Figure 10 and 11 are fixed.
Reviewer#2 - Concern # 7: English Writing: This paper requires extensive proofreading. There are some of grammatical, spelling and typos problems. The authors have to thoroughly scrutinize the paper. Without professional, accurate and clear English, readers cannot understand the research.
Author response: Thanks for the recommendation. Paper is now carefully checked for English Grammar and typos etc.
Author action: Corrections are done at various places throughout the paper.
Reviewer#2 - Concern # 8: References list: References should follow the MDPI-Sensors style. For example, Researchers' names should follow the Sensors Journal style (please check all the references). The double quotation should be removed from all reference names. Some search names in the reference list begin an uppercase letter for each word (such as [3], [6] ... etc.) and others use only an uppercase letter in the first word (such as [1], [2] … etc.), the author should standardize style. The number of references is insufficient for this study. The references list requires extensive scrutiny by the authors.
Author response: Thanks for the recommendation.
Author action: All references are now changed to MDPI Sensor Style.
We have thoroughly checked the paper text to improve its readability and English language. Several spellings and many grammatical corrections are made in the paper text. We hope that the paper text is in much better shape now.
Please contact me if you have any questions about the revised manuscript.
Best regards,
Muhammad Zia Ul Haq et al.
Round 2
Reviewer 1 Report
One of my biggest concerns was the lack of performance comparison with recent schemes as I listed in the previous review report. Unfortunately, the authors fail to revise the manuscript according to this comment. So, I don't agree with the acceptance of this manuscript for publication.
Reviewer 2 Report
The authors have responded to some comments but there are still some comments that require a response. However, the authors have to address accurately all of the below comments.
- Literature Review Section: This comment still requires not response. The word "this" (page3-line125) is not clear, is it related to the research [7] or to the proposed research? It can be replaced with “their”. We believe the sentences from "analysis, we suggest that the following design important criteria that should" (page6-line239) to "solutions should use it to ensure network connectivity." (page6-line253) should be moved to another section such as Introduction Section.
- Simulation Results and Analysis Section: The authors adequately explained the results and trials but did not discuss these results in a separate section. The authors compared their results with DPV and ADPV but did not compare their results with existing research.
- Figures and Tables: Figure 4 used in-text before Figure 3. What is the point of using some colors in tables and figures? Why not rely on one color and one format to represent the proposed research in Figures 10 and 11? This comment still requires a response.
- English Writing: This paper requires minor proofreading. There are some of grammatical, spelling and typos problems. The authors have to thoroughly scrutinize the paper. Without professional, accurate and clear English, readers cannot understand the research.
- References list: References should follow the MDPI-Sensors style. For example, Researchers' names should follow the Sensors Journal style (please check all the references). Double quotation should be removed from all reference names. Some search names in the reference list begin an uppercase letter for each word (such as [3], [6] ... etc.) and others use only an uppercase letter in the first word (such as [1], [2] … etc.), author should standardize style. The references list requires extensive scrutiny by the authors.